# Pharmacological Inhibition of Astrocytic Transglutaminase 2 Facilitates the Expression of a Neurosupportive Astrocyte Reactive Phenotype in Association with Increased Histone Acetylation

**DOI:** 10.3390/biom14121594

**Published:** 2024-12-13

**Authors:** Thomas Delgado, Jacen Emerson, Matthew Hong, Jeffrey W. Keillor, Gail V. W. Johnson

**Affiliations:** 1Department of Anesthesiology and Perioperative Medicine, University of Rochester, 601 Elmwood Ave, Box 604, Rochester, NY 14620, USA; thomas_delgado@urmc.rochester.edu (T.D.); jacene@umich.edu (J.E.); mhong12@u.rochester.edu (M.H.); 2Department of Chemistry and Biomolecular Sciences, University of Ottawa, Ottawa, ON K1N6N5, Canada; jkeillor@uottawa.ca

**Keywords:** astrocytes, transcriptional regulation, lipid metabolism, neurite outgrowth, proteomics

## Abstract

Astrocytes play critical roles in supporting structural and metabolic homeostasis in the central nervous system (CNS). CNS injury leads to the development of a range of reactive phenotypes in astrocytes whose molecular determinants are poorly understood. Finding ways to modulate astrocytic injury responses and leverage a pro-recovery phenotype holds promise in treating CNS injury. Recently, it has been demonstrated that ablation of astrocytic transglutaminase 2 (TG2) shifts reactive astrocytes towards a phenotype that improves neuronal injury outcomes both in vitro and in vivo. Additionally, in an in vivo mouse model, pharmacological inhibition of TG2 with the irreversible inhibitor VA4 phenocopied the neurosupportive effects of TG2 deletion in astrocytes. In this study, we extended our comparisons of VA4 treatment and TG2 deletion to provide insights into the mechanisms by which TG2 attenuates neurosupportive astrocytic function after injury. Using a neuron–astrocyte co-culture model, we found that VA4 treatment improves the ability of astrocytes to support neurite outgrowth on an injury-relevant matrix, as we previously showed for astrocytic TG2 deletion. We hypothesize that TG2 mediates its influence on astrocytic phenotype through transcriptional regulation, and our previous RNA sequencing suggests that TG2 is primarily transcriptionally repressive in astrocytes, although it can facilitate both up- and downregulation of gene expression. Therefore, we asked whether VA4 inhibition could alter TG2’s interaction with Zbtb7a, a transcription factor that we previously identified as a functionally relevant TG2 nuclear interactor. We found that VA4 significantly decreased the interaction of TG2 and Zbtb7a. Additionally, we assessed the effect of TG2 deletion and VA4 treatment on transcriptionally permissive histone acetylation and found significantly greater acetylation in both experimental groups. Consistent with these findings, our present proteomic analysis further supports the predominant transcriptionally repressive role of TG2 in astrocytes. Our proteomic data additionally unveiled pronounced changes in lipid and antioxidant metabolism in astrocytes with TG2 deletion or inhibition, which likely contribute to the enhanced neurosupportive function of these astrocytes.

## 1. Introduction

Astrocytes are unique and versatile cells of the central nervous system (CNS). In homeostatic conditions, astrocytes maintain the blood–brain barrier, regulate the extracellular matrix (ECM), and provide crucial metabolic support for neurons, among other functions [1,2,3]. Inflammatory and stress signals, whether from injury, infection, or other sources, cause astrocytes to take on a range of phenotypes from neuroprotective to neurotoxic, although the exact molecular determinants of these phenotypes are poorly understood [4,5,6,7,8,9,10,11,12]. Our lab has previously identified the protein transglutaminase 2 (TG2) as a key factor in modulating the phenotype of reactive astrocytes [13,14,15,16,17].

TG2 is a widely expressed, multi-functional protein in the transglutaminase family and is the most highly expressed transglutaminase in the brain. TG2 has been studied mostly as a transamidating enzyme, which catalyzes the incorporation of an amine into a glutamine of the acceptor protein [18,19,20,21,22]. This reaction requires calcium which stabilizes TG2 in a catalytically active, open conformation [23,24]. Conversely, GTP/GDP stabilizes TG2 in a closed, catalytically inactive conformation which, due to the relatively high intracellular concentrations of GTP/GDP compared to calcium, is the predominant intracellular conformation of TG2 [25,26]. In the closed conformation, TG2 can function as a scaffold for protein–protein interactions at the cell surface and in the nucleus, with well-described roles in cell–ECM interactions and chromatin regulation [18,20,21,25,27,28,29,30]. Additionally, the TG2 promoter contains transcription factor binding sites associated with inflammation and hypoxia; therefore, TG2 expression is often greatly increased in injury conditions, allowing TG2 to have significant influence on cellular injury responses through its transamidating or scaffolding roles [31,32,33].

We have previously shown that astrocytic TG2 greatly influences neuronal survival and recovery in multiple injury models. TG2 deletion makes astrocytes more resilient to oxygen–glucose deprivation and improves their ability to protect neurons in these conditions [13,17,34]. In vivo, mice with TG2 knocked out of astrocytes show significantly faster motor function recovery after spinal cord injury (SCI) compared to wild type (WT) mice [14]. To isolate potential mechanisms underlying this intriguing finding, we modeled axonal regeneration in vitro using a neurite outgrowth assay in which we grew neurons on an injury-relevant, growth-inhibitory matrix comprised of chondroitin sulfate proteoglycans (CSPGs) and paired them with TG2 knockout (TG2−/−) or WT astrocytes. CSPGs are an important component of the post-SCI environment as they are densely deposited in and around the lesion core and inhibit axonal regeneration across the lesion [11,35,36]. We found that TG2−/− astrocytes facilitated neurite outgrowth across the simulated lesion matrix to a greater extent than WT astrocytes [15]. The mechanisms through which TG2−/− astrocytes can better promote neurite outgrowth on a growth-inhibitory matrix likely also contribute to the enhanced functional recovery of astrocytic TG2−/− mice after SCI; however, the exact molecular and functional aspects of these astrocytic changes have yet to be identified.

We hypothesize that TG2 attenuates neurosupportive functional adaptations in astrocytes under stress and that this effect is mediated by its ability to facilitate transcriptional repression of genes associated with these functions. In response to stress, TG2 can migrate in or out of the nucleus, depending on the cell type [34,37]. Within the nucleus, TG2 can interact with a range of transcription factors, chromatin regulatory proteins, and histones to regulate gene expression [15,16,38,39,40,41]. Therefore, TG2 likely plays a multi-faceted role in epigenetic regulation. Recently, the Maze lab found that TG2 can serotonylate and dopaminylate histones, which is primarily associated with promoting gene expression [21,22]. Yet, TG2 deletion in astrocytes is associated with a predominant upregulation of genes, and RNA sequencing of TG2−/− astrocyte cultures has shown about a 2/3 upregulation and 1/3 downregulation of genes [17]. This suggests, in astrocytes, that TG2 has nuanced roles in control of gene expression, but, overall, it is primarily transcriptionally repressive.

To probe the gene-regulatory mechanisms underlying the influence that TG2 has on astrocytic function during stress, we inhibited TG2 using our small molecule inhibitor VA4 [42]. VA4 irreversibly modifies TG2 function by binding and blocking active site residues necessary for its catalytic activity and “locking” TG2 in its open conformation, thereby also potentially altering protein scaffolding function associated with its closed conformation [23,26,43,44,45]. Previously, we demonstrated that VA4 treatment of WT astrocytes phenocopied functional effects of astrocytic TG2 deletion. In vitro, both TG2 deletion and VA4 treatment significantly improved astrocyte survival after an ischemic insult [16], while in vivo, VA4 treatment of WT mice significantly improved motor function recovery after SCI, compared to vehicle treatment, to an extent similar to that observed in mice with astrocyte-specific TG2 deletion [14]. These data indicate that VA4 treatment, like TG2 deletion, improves the resilience of astrocytes in stress conditions and promotes a neurosupportive phenotype in reactive astrocytes after CNS injury. In the present study, we extended our use of VA4 as a comparative tool to find novel transcriptionally permissive changes associated with TG2 deletion. Additionally, we used proteomic analysis to identify differentially regulated pathways that may underlie the phenotypic changes of TG2−/− astrocytes associated with enhanced neuronal support during stress.

## 2. Methods and Materials

### 2.1. Animals

All mice and rats were maintained in a 12 h light/dark cycle with food and water available ad libitum. The animal study protocol was approved by the University Committee on Animal Resources of the University of Rochester (protocol #102079/2007-023E, approved 14 May 2021). WT C57BL/6 mice were originally purchased from Charles River Laboratories. Our TG2−/− mice on a C57Bl/6 background were described previously and have been continuously bred in house [13]. Timed-pregnant Sprague Dawley rats were obtained from Charles River Laboratories.

### 2.2. Cell Culture

Primary astrocytes were cultured between post-natal days 0 to 1 (P0–P1) from either WT C57BL/6 or TG2−/− mouse pups as previously described [13]. In brief, P0–P1 mouse pups were rapidly decapitated, the brains were collected and dissected, meninges removed, and cortical hemispheres were collected. Following trituration, cells were plated onto culture dishes in MEM supplemented with 10% FBS (Atlas Biologicals, Fort Collins, CO, USA, F-0500-DR), 33 mM glucose, 1 mM sodium pyruvate (Gibco, Grand Island, NY, USA, 11360-070), and 0.2% Primocin (Invivogen, San Diego, CA, USA, ant-pm-05) (glial MEM). Cortical tissues were pooled from all pups in a litter during plating, so our astrocyte cultures were mixed sex. Twenty-four hours after plating, the dishes were shaken vigorously and rinsed to remove debris and other cell types. Astrocytes were maintained at 37 °C/5% CO_2_ for 7–8 days, frozen in glial MEM containing 10% DMSO, and stored in liquid nitrogen. For experiments, astrocytes were thawed, grown, and passaged in glial MEM, and only cultures on 2nd or 3rd passages at no greater than 90% confluency were used for final data acquisition.

Primary cortical neurons were prepared from Sprague Dawley rat embryos at embryonic day 18 (E18) and cultured as previously described with some modifications [46]. To prepare the coverslips/wells, poly-D-lysine (Sigma, St. Louis, MO, USA, P6407) was diluted in PBS to a concentration of 20 µg/mL and added to the wells for 4 h. The wells were either rinsed and stored with PBS, or after rinsing, CSPGs (Millipore, Burlington, MA, USA, CC117) in PBS (2.5 µg/mL) were added and incubated overnight to coat the coverslips. All wells and coverslips were rinsed with PBS prior to plating the neurons. To prepare the neurons, pregnant rats were euthanized using CO_2_, followed by rapid decapitation in accordance with NIH Animal Research Advisory Committee guidelines. Embryos were removed, rapidly decapitated, brains were extracted, cerebral cortices dissected, and meninges were removed. Cerebral cortices were then digested in trypsin-EDTA (0.05%) (Corning, Kennebunk, ME, USA, 25-053-Cl) for 15–20 min in a 37 °C water bath. Following gentle trituration, neurons were plated in neuron plating medium consisting of MEM (Gibco, 42360032) supplemented with 5% FBS, 20 mM glucose, and 0.2% Primocin at a density of 12,000 cells/cm^2^ on the coated coverslips. Cortical tissues were pooled from all pups in a litter during plating, so our neuron cultures were mixed sex. Four to five hours after plating, the medium was replaced with Neurobasal medium (Gibco, 21103-049) containing 2% B27 (Gibco, 17504-044), 0.5 mM Glutamax (Gibco, 35050-061), and 0.2% Primocin (neuron growth medium). Neurons were incubated at 37 °C/5% CO_2_ and experiments begun at days in vitro (DIV) 1.

HEK 293TN cells (System Biosciences, Palo Alto, CA, USA, LV900A-1) were thawed and grown at 37 °C/5% CO_2_ in DMEM (Gibco 11995-065) supplemented with 10% FBS, 1× GlutaMAX (Gibco 35050-061), and 4.5 µg/mL gentamicin (Gibco 15710-064). Cells were grown to confluency and passaged using trypsin-EDTA (0.05%) (Corning, 25-053-Cl) at least once before use in immunoprecipitation (IP) experiments.

### 2.3. VA4 Treatment

Cultured cells were treated with 10 µM VA4 or 0.04% DMSO for vehicle control. For IP experiments, HEK 293TN cells were treated with VA4 12 h after transfection (48 h before protein collection). For immunoblot experiments, astrocytes were treated with VA4 for 48 h before protein collection. For neurite outgrowth studies, astrocytes were pre-treated with VA4 for 4 days before being paired with neurons. The astrocyte cultures received a complete medium change and replacement of VA4 after 2 days of incubation. After pairing with DIV 1 neurons, astrocytes were treated with VA4 throughout the experiment, from DIV 1 to DIV 5 of neuron culture, receiving a complete medium change and VA4 replacement at DIV 3.

### 2.4. Neurite Outgrowth Analyses

WT and TG2−/− astrocytes, treated with VA4 or DMSO vehicle, were seeded onto transwell inserts (6.5 mm) with a membrane pore size of 1.0 µm (Grenier Bio-One, Monroe, NC, USA, 665610) in glial MEM 48 h prior to pairing with the neurons. The astrocyte transwell cultures were switched into supplemented neurobasal medium 24 h before pairing. On neuron DIV 1, the inserts were placed over the neuron coverslips of a 12-well plate and the neurons received a half medium change with astrocyte-conditioned neurobasal medium. The cell pairs were incubated for 96 h, and neuron coverslips were collected for analysis of neurite outgrowth.

Neuron coverslips from the transwell co-cultures were washed three times with PBS, followed by fixation with 2% paraformaldehyde and 4% sucrose in PBS for 5 min. After three washes with PBS, the cells were permeabilized with 0.25% Triton X-100 in PBS and blocked with PBS containing 5% BSA and 0.3 M glycine. Rabbit anti-MAP2 (Cell Signaling, Danvers, MA, USA, #8707S) was diluted in blocking buffer (1:200) and incubated overnight on the coverslips. The next day, the coverslips were washed three times and incubated in Alexa Fluor 594 donkey anti-rabbit (Invitrogen, Frederick, MD, USA, A21207) for 1 h. Coverslips were then counterstained with Hoechst 33,342 (1:10,000) and mounted using Fluoro-gel in TES Buffer (Electron Microscopy Sciences, Hatfield, PA, USA, 17985-30). The slides were imaged using a Zeiss Observer D1 microscope (Zeiss, Oberkochen, Germany) with a 40× objective.

Ten to fifteen neurons per coverslip were imaged for each experimental group. Images were processed by ImageJ Fiji (version 2.14.0/1.54f) using the Simple Neurite Tracer (SNT) plugin. Neurites were traced using a scale of 6.7 pixel/µm. For the max neurite length studies, the longest neurite of each neuron was recorded.

### 2.5. Constructs

The use of V5-tagged human TG2 in pcDNA and in the lentiviral vector FigB has been described previously [46,47]. The FLAG-tagged Zbtb7a construct was purchased from Origene (Pleasant View, UT, USA, RC222759).

### 2.6. Immunoblotting

For protein collection, cells were lysed and collected in IP lysis buffer (150 mM NaCl, 50 mM Tris-HCl, 1 mM EDTA, 1 mM EGTA, 0.5% NP-40 in PBS). Protein concentrations were measured using a BCA assay. Samples were prepared at 1 µg/1 µL in 1× SDS containing IP lysis buffer and boiled at 100 °C for 10 min. Protein samples were resolved on 12% SDS-PAGE gels and proteins transferred to a nitrocellulose or PVDF membrane. Membranes were blocked in blocking buffer, 5% milk in Tris-buffered saline with Tween 20 (TBS-T) (20 mM Tris base, 137 mM NaCl, 0.05% Tween 20), for 1 h at room temperature. After blocking, primary antibodies against FLAG tag (CST 8146S), V5 tag (CST 13202S), GAPDH (Proteintech, Rosemont, IL, USA, 60004-1-Ig), acetylated histone H3 K9 (CST 9649S), or beta tubulin (rabbit polyclonal antibody, Proteintech 10094-1-AP) were added to the blots in fresh blocking buffer and incubated at 4 °C overnight. The next day blots were washed with TBS-T and incubated for 1 h at room temperature with HRP-conjugated secondary antibody in blocking buffer. The blots were washed with TBS-T before being visualized with an enhanced chemiluminescence reaction. ImageJ Fiji (version 2.14.0/1.54f) was used to quantify the intensity of each band, and all values were normalized to GAPDH or beta tubulin levels.

### 2.7. Co-Immunoprecipitation

For immunoprecipitation (IP), HEK 293TN was transfected with V5-TG2 and FLAG-Zbtb7a constructs using PolyJet transfection reagent (Signagen, Frederick, MD, USA, SL100688) following the manufacturer’s protocol. The cells were treated with VA4 12 h after transfection and collected 48 h after VA4 treatment. The cells were lysed and collected in IP lysis buffer (150 mM NaCl, 50 mM Tris-HCl, 1 mM EDTA, 1 mM EGTA, 0.5% NP-40 in PBS). Protein concentrations were measured using a BCA assay. A 300 µg aliquot of lysate was immunoprecipitated with 4 µL of rabbit anti-V5 tag antibody (CST 13202S). After addition of primary antibody, the samples were incubated on a rotator at 4 °C overnight. IgG control samples were incubated with an equivalent amount of normal rabbit (Millipore 12-370) IgG antibody. After 18 h, 30 µL of Pierce protein A/G magnetic agarose beads, (Thermo Scientific, Waltham, MA, USA, 78609) washed in IP wash buffer (2 mM EDTA, 0.1% NP-40 in PBS) and blocked in 1% BSA in PBS, were added. After a 4 h incubation, rotating at 4 °C, the samples were thoroughly washed in IP wash buffer and then in IP lysis buffer. After washing, beads were incubated in 30 µL of 2.5× SDS in IP lysis buffer for 10 min at 100 °C. Samples were then immunoblotted as described above. For quantification, ImageJ Fiji (version 2.14.0/1.54f) was used to measure the intensity of FLAG and V5 bands in each lane and the FLAG signal was normalized to that of V5.

### 2.8. Sample Preparation for Liquid Chromatography with Tandem Mass Spectrometry (LC-MS/MS)

#### 2.8.1. Cell Culture for Mass Spectrometry

Astrocytes were cultured in 60-mm dishes for 7 days in their second passage, with a half-medium change every 3 days. They were then washed with PBS, trypsinized, washed again, and frozen as cell pellets prior to further analysis. For VA4 experiments, WT astrocytes were cultured in 60-mm dishes for 7 days in their second passage, with a half-medium change on DIV 3. On DIV 5, they received a full medium change with 10 µM VA4 or 0.04% DMSO. On DIV 7, they were then processed as above and frozen as cell pellets. Cell pellets were submitted to the URMC Mass Spectrometry Core.

#### 2.8.2. Sample Preparation

Cell lysis was performed by adding 300 µL of 5% SDS, 100 mM triethylammonium bicarbonate (TEAB) per 6 × 10^6^ cells. Samples were vortexed and then sonicated (QSonica, Newtown, CT, USA) for 5 cycles, with a 1 min resting period on ice after each cycle. Lysates were then centrifuged at 15,000× *g* for 5 min to collect cellular debris, and the supernatant was collected. Protein concentration was determined by BCA (Thermo Scientific), after which samples were diluted to 1 mg/mL in 5% SDS, 50 mM TEAB.

Twenty-five micrograms of protein from each sample were reduced with 2 mM dithiothreitol, followed by incubation at 55 °C for 60 min. Iodoacetamide was added to 10 mM and incubated in the dark at room temperature for 30 min to alkylate the proteins. Phosphoric acid was added to 1.2%, followed by six volumes of 90% methanol, 100 mM TEAB. The resulting solution was added to S-Trap micros (Protifi, Long Island, NY, USA) and centrifuged at 4000× *g* for 1 min. The S-Traps containing trapped protein were washed twice by centrifuging through 90% methanol, 100 mM TEAB. A total of 1 µg of trypsin was brought up in 20 µL of 100 mM TEAB and added to the S-Trap, followed by an additional 20 µL of TEAB to ensure the sample did not dry out. The cap to the S-Trap was loosely screwed on but not tightened to ensure the solution was not pushed out of the S-Trap during digestion. Samples were placed in a humidity chamber at 37 °C overnight. The next morning, the S-Trap was centrifuged at 4000× *g* for 1 min to collect the digested peptides. Sequential additions of 0.1% trifluoroacetic acid (TFA) in acetonitrile (ACN) and 0.1% TFA in 50% ACN were added to the S-Trap, centrifuged, and pooled. Samples were frozen and dried down in a speed vac (Labconco, Kansas City, MO, USA), then re-suspended in 0.1% trifluoroacetic acid prior to analysis.

Samples related to Figure 5a,b,e were reconstituted in TEAB to 4 mg/mL in 100 mM TEAB, then labeled with tandem mass tag (TMT) 10 plex reagents (Thermo Scientific) following the manufacturer’s protocol. Briefly, TMT tags were removed from the −20 °C freezer and allowed to equilibrate to RT for 5 min, after which, 22 µL of ACN was added to each tag. In total, 20 µL of individual TMT tags was added to respective samples and the reactions were carried out at RT for 1 h, after which, the reaction was quenched by adding 5% hydroxylamine. All 10 samples were combined and dried down in a speed vac prior to high-pH fractionation.

Labeled peptides were fractionated using homemade C18 spin columns. The C18 was activated by two 50 µL washes of ACN via centrifugation followed by equilibration by two 50 µL washes of 10 mM ammonium hydroxide (NH_4_OH). Peptides were resuspended in 50 µL of 10 mM NH_4_OH and added to the spin column. After centrifugation, the column was washed twice with 10 mM NH_4_OH. Fractions were eluted off the column with centrifugation by stepwise addition of 10 mM NH_4_OH with increasing percentages of ACN as follows: 2, 3.5, 5, 6.5, 8, 9.5, 11, 12.5, 14, 15.5, 17, 18.5, 20, 21.5, 27, 50%. The 16 fractions were concatenated down to 8 by combining fractions 1 and 9, 2 and 10, 3 and 11, etc. Fractionated samples were frozen, dried down in the speed vac, and brought up in 0.1% TFA prior to mass spectrometry analysis.

### 2.9. LC-MS/MS

#### 2.9.1. Data Collection

Peptides were injected onto a homemade 30 cm C18 column with 1.8 um beads (Sepax, Newark, DE, USA), with an Easy nLC-1200 HPLC (Thermo Fisher), connected to a Fusion Lumos Tribrid mass spectrometer (Thermo Fisher). Solvent A was 0.1% formic acid (FA) in water, while solvent B was 0.1% FA in 80% ACN. Ions were introduced to the mass spectrometer using a Nanospray Flex source operating at 2 kV. Data were collected by two different methods: data-dependent acquisition (DDA) with TMT-MS3 and data-independent acquisition (DIA).

For the data collected with DDA TMT-MS3 (see Appendix A), the gradient began at 3% B for 2 min, increased to 10% B over 7 min, increased to 38% B over 94 min, then ramped up to 90% B over 5 min where it was held at 90% B for 3 min before returning to 0% B for 2 min and re-equilibrating for 7 min, for a total runtime of 120 min. The Fusion Lumos was operated in data-dependent mode, with MultiNotch Synchronized Precursor Selection MS3 (SPS-MS3) enabled to increase quantitative accuracy [48]. The cycle time was set to 3 s with monoisotopic precursor selection set to ‘Peptide’. MS1 scans were acquired in the Orbitrap at a resolution of 120,000 at *m*/*z* of 200 over a range of 400–1500 *m*/*z*, with an AGC target of 4 × 10^5^, and a maximum ion injection time of 50 ms. Precursor ions with a charge state of 2–5 were selected for fragmentation by collision-induced dissociation (CID) using a collision energy of 35% and an isolation width of 1.0 *m*/*z*. MS2 scans were acquired in the ion trap with an AGC target of 1 × 10^4^ and a maximum ion injection time of 35 ms. Dynamic exclusion was set to filter precursor ions after 1 time with a duration of 45 s and high and low mass tolerances of 10 ppm using a maximum intensity threshold of 1 × 10^20^ and minimum intensity threshold of 1 × 10^4^. MS3 scans were performed by selecting the 10 most intense fragment ions between 400–200 *m*/*z* with an isolation width of 2 Da, excluding any ions that were 40 *m*/*z* less or 10 *m*/*z* greater than the precursor ions, which were then fragmented using higher energy collisional dissociation (HCD) using a collision energy of 60%. MS3 ions were detected in the Orbitrap with a resolution of 50,000 at *m*/*z* 200 over a range of 100–300 *m*/*z* with an AGC target of 1 × 10^5^, a normalized AGC target of 200%, and a maximum ion injection time of 100 ms.

For the data collected with data-independent acquisition (DIA) (Figure 5c,d,f) (see Appendix A), the gradient began at 4% B for 2 min, increased to 28% B over 66 min, increased to 38% over 7 min, increased to 90% B over 5 min, and was held at 90% B for 3 min to wash the column, before returning to 0% B over 2 min and re-equilibrating for 5 min, for a total runtime of 90 min. The Fusion Lumos was operated in data-independent acquisition (DIA) mode, with MS1 scans acquired in the Orbitrap at a resolution of 60,000 over a range of 395–1005 *m*/*z*, with an AGC target of 4 × 10^5^, and a maximum injection time of 50 ms. MS2 scans were acquired in Orbitrap with a resolution of 15,000 over a range of 200–2000 *m*/*z*, with a maximum ion injection time of 23 ms, HCD collision energy set to 33%, an AGC target of 4 × 10^5^, and normalized AGC of 800%. Precursor ions were sampled using a staggered windowing scheme of 8 *m*/*z* with 4 *m*/*z* overlaps for a total of 75 windows between MS1 scans.

#### 2.9.2. Data Analysis

For the data collected with DDA TMT-MS3, raw data were searched using the SEQUST search engine within the Proteome Discoverer platform, version 2.4 (Thermo Scientific), using the UniProt Mouse database (downloaded 27 April 2021). Trypsin was selected as the enzyme allowing up to 2 missed cleavages, with an MS1 mass tolerance of 10 ppm, and an MS2 mass tolerance of 0.6 Da. Carbamidomethylation of cysteine and TMT on lysine and peptide N-terminus were set as fixed modifications. Oxidation of methionine was set as a variable modification. Percolator was used as the FDR calculator, filtering out peptides with a q-value of greater than 0.01. Reporter ions were quantified using the ‘Report Ions Quantifier’ node with an integration tolerance of 20 ppm and integration method set to ‘most confident centroid’. Protein abundances were calculated by summing the signal to noise of the report ions from each identified peptide. *p* values were measures using Student’s *t*-test within Proteome Discoverer.

For the data collected with DIA, raw data were searched with DIA-NN version 1.8.1 (https://github.com/vdemichev/DiaNN) [49]. For all experiments, data analysis was carried out using library-free analysis mode in DIA-NN. To annotate the library, the mouse UniProt ‘one protein sequence per gene’ database (UP000000589_10090, downloaded 12 September 2021) was used with ‘deep learning-based spectra and RT prediction’ enabled. For precursor ion generation, the maximum number of missed cleavages was set to 1, cysteine carbamidomethylation set as a fixed modification, maximum number of variable modifications to 1 for Ox(M), peptide length range to 7–30, precursor charge range to 2–3, precursor *m*/*z* range to 400–1000, and fragment *m*/*z* range to 200–2000. The quantification was set to ‘Robust LC (high precision)’ mode with normalization set to RT-dependent, MBR enabled, protein inferences set to ‘Genes’, and ‘Heuristic protein inference’ turned off. MS1 and MS2 mass tolerances, along with the scan window size were automatically set by the software. Precursors were subsequently filtered at library precursor q-value (1%), library protein group q-value (1%), and posterior error probability (20%). Protein quantification was carried out using the MaxLFQ algorithm as implemented in the DIA-NN R package (https://github.com/vdemichev/diann-rpackage) and the number of peptides quantified in each protein group was counted as implemented in the DiannReportGenerator R Package (https://github.com/URMC-MSRL/DiannReportGenerator) [50]. Downstream processing and statistical analyses were performed using the Perseus software [51]. Specifically, proteins IDs were filtered to only allow proteins identified with 2 or more peptides in at least two samples in one biological group. Missing values were then imputed from a normal distribution with a standard deviation of 0.3 and a downshift of 1.8, and a two-sample Student’s *t*-test was performed on the imputed data. Perseus output was converted to excel format using the ProteinReportr R package (https://github.com/URMC-MSRL/ProteinReportr).

### 2.10. Protein Differential Expression Analysis

Differentially expressed proteins were identified using pairwise *t*-tests followed by Benjamini–Hochberg (BH) FDR correction. Data were sorted by log2 fold change and adjusted *p* value. Log2 fold change values between 0.5 and −0.5 and adjusted *p* values > 0.05 were excluded from further analysis. Differential expression was presented in volcano plots, which were generated using GraphPad Prism 10. A Venn diagram was generated using BioVenn [52].

### 2.11. Gene Ontology Analyses

Sorted differentially expressed gene symbol lists were analyzed using DAVID GO Biological Process [53]. The enrichment data were extracted for generating dot plots using ggplot2 package in R 4.3.2. Gene lists were also analyzed by Enrichr to yield consensus transcription factor enrichments based on ENCODE and ChEA databases of multi-omic, chromatin immunoprecipitation data [54,55,56].

### 2.12. Statistical Analysis

GraphPad Prism 10 was used to report the raw data and perform statistical analysis. Outliers were evaluated using ROUT with Q = 1% for all figures. Normality was tested using a Shapiro–Wilk test. The mean and standard error of the mean were calculated for each group. For significance testing, a Kruskal Wallis test was used in Figure 1 and unpaired *t*-tests were used in Figures 2–4, and levels of significance were set at * *p* < 0.05, ** *p* < 0.01, *** *p* < 0.001, **** *p* < 0.0001. Details of statistical analyses are given in Appendix A.

## 3. Results

### 3.1. VA4 Neurite Outgrowth

Recently, we demonstrated that TG2 deletion in astrocytes enhances their ability to support neurite outgrowth on a growth-inhibitory matrix, which simulates the extracellular matrix of an SCI lesion core [15]. This finding provides potentially important insight into the mechanisms underlying the enhanced functional recovery of astrocytic TG2 deleted mice after SCI [14]. Notably, we could phenocopy this effect by treating mice with VA4 after SCI [14]. Therefore, we asked whether TG2 inhibition by VA4 treatment of astrocyte cultures could replicate our previous neurite outgrowth data with TG2−/− astrocytes. For these experiments, WT astrocytes were grown on transwells and treated with VA4 or DMSO vehicle. They were then paired with neurons seeded on a growth-inhibitory matrix comprised of CSPGs on DIV 1 of neuron culture (Figure 1a). As expected, the presence of WT astrocytes (either VA4-treated or vehicle-treated) resulted in significantly greater neurite outgrowth on CSPGs compared to unpaired neurons. Similar to the enhanced neurite outgrowth observed in the presence of TG2−/− astrocytes [15], neurons paired with VA4-treated astrocytes promoted significantly greater neurite outgrowth compared to vehicle-treated astrocytes (Figure 1b,c). Importantly, VA4 treatment of neurons alone did not affect their neurite outgrowth on a growth-supportive poly D Lysine (PDL) or CSPG matrix (Appendix A), and VA4 treatment of TG2−/− astrocytes showed no added effect on their ability to support neurite outgrowth on a CSPG matrix (Appendix A).

### 3.2. Immunoprecipitation of TG2 and Zbtb7a After VA4 Treatment

The ability of TG2 to regulate gene expression has been well-described in the literature and our previous data suggest that this role of TG2 is integral to its ability to influence injury response across cell types [14,17,18,28,38,39,40,41,47,57,58,59,60,61]. We hypothesize that TG2 is an important regulator of neurosupportive gene expression in reactive astrocytes; however, the mechanisms by which it regulates gene expression and the functionally significant genes impacted in astrocytes have not been fully delineated. We previously posited that the interaction between TG2 and Zbtb7a, a transcription factor that is ubiquitous throughout the nucleus, is an important contributor to the ability of TG2 to regulate gene expression in reactive astrocytes [15]. Having established the ability of VA4 treatment to phenocopy the effects of TG2 deletion in our neurite outgrowth model, we asked whether VA4 could alter the interaction between TG2 and Zbtb7a. To address this question, HEK cells were transfected with V5-tagged TG2 and FLAG-tagged Zbtb7a, followed by VA4 or vehicle-only treatment [15]. IP of TG2 from the VA4-treated cell lysates resulted in significantly less pull down of Zbtb7a compared to vehicle-only treated cells (Figure 2), demonstrating reduced interaction between these proteins.

### 3.3. Histone Acetylation in WT and TG2−/− Astrocytes

Previous data from our lab suggest that TG2 has both a minor activating effect and a predominant repressive effect on gene expression in astrocytes [14,17], and we hypothesize that interactions with Zbtb7a and other epigenetic regulators facilitate TG2’s transcriptional effects. Concordantly, Zbtb7a has been independently shown to facilitate both transcriptional activation and repression in association with its ability to promote accessibility of transcription factors to gene promoters and its ability to bind to repressive chromatin regulatory complexes [62]. Previous yeast two-hybrid and interactome analyses revealed that TG2 and Zbtb7a both interact with components of Sin3a, a transcriptional repressor complex that facilitates chromatin modifications by HDAC1 and HDAC2 [15,63,64,65]. Notably, the Sin3a complex is an important regulator of injury responses across cell types [37,66,67]. With these data in mind, we hypothesized that TG2 limits the chromatin accessibility of neurosupportive genes in reactive astrocytes in part through regulating histone acetylation. Acetylated H3K9 (H3K9ac) localizes to active gene promoters and is associated with enhanced transcriptional activity [68]. Therefore, we measured global H3K9ac levels in WT and TG2−/− astrocytes and found that TG2−/− astrocytes had significantly higher acetylation levels than WT astrocytes at steady-state in glial media (Figure 3).

### 3.4. Histone Acetylation After VA4 Treatment in WT Astrocytes

Given the significantly higher levels of H3K9ac in TG2−/− astrocytes, we asked whether VA4 treatment of WT astrocytes could replicate this epigenetic effect. VA4-treated astrocytes exhibited a significant increase in H3K9ac levels compared to vehicle-treated astrocytes (Figure 4), comparable to the levels observed in TG2−/− astrocytes (Figure 3).

### 3.5. Differential Proteomic Analysis in TG2−/− and VA4-Treated Astrocytes

Our lab has previously analyzed transcriptional differences between TG2/− and WT astrocyte cultures using RNA sequencing [17]. To extend these findings and provide more insight into the functional differences observed in astrocytes with TG2−/− deletion or inhibition that may explain their neurosupportive phenotype, we analyzed the proteome of TG2−/− astrocytes and VA4-treated astrocytes in steady-state conditions (Figure 5). Differential analyses comparing TG2−/− and VA4-treated astrocytes to WT and vehicle-treated astrocytes, respectively, show strong enrichments in lipid metabolic pathways through GO Biological Process (Figure 5b,d). Additionally, both analyses show top enrichments of genes associated with NFE2L2, or NRF2, as well as Zbtb7a, through transcription factor gene ontology categories (Figure 5e,f). In comparison to wild type astrocytes, VA4-treated astrocytes show a wider range of differentially regulated proteins than the TG2−/− astrocytes; however, TG2−/− astrocytes share the majority, about 70%, of their differentially regulated proteins with VA4-treated astrocytes (Figure 5g). Replicate proteomic analysis of TG2−/− and WT astrocytes using data independent acquisition (DIA) is shown in Appendix A. All data from proteomic analyses are included as Appendix A.

## 4. Discussion

Herein, we show that VA4 treatment of WT astrocytes phenocopies the ability of TG2−/− astrocytes to promote greater neurite outgrowth on a growth-inhibitory CSPG matrix compared to untreated WT astrocytes. The implications and limitations of our neurite outgrowth findings were discussed in depth in our previous paper [15]. Chiefly, our neurite outgrowth data suggest that TG2 deletion or inhibition in astrocytes improves their ability to support highly energy-dependent processes in neurons in a stress context. CSPGs have been extensively studied in the context of SCI, and intriguing data suggest that their growth-inhibitory effect on neurons may in part be due to inhibition of autophagic processes, which may induce energetic and proteostatic stress [69]. Astrocytes can respond to neuronal stress cues by upregulating neurosupportive functions, and this response is likely altered in astrocytes in which TG2 is deleted or inhibited. The underlying astrocytic functional differences that contribute to our neurite outgrowth findings may also contribute to the improved motor function recovery after SCI observed in VA4-treated and astrocytic TG2−/− mice by enhancing axonal regeneration [15]. This study provides insight into the mechanisms through which TG2 acts to influence astrocytic response to stress and which astrocytic functions are predominantly affected by TG2 deletion or inhibition.

TG2 is an injury-responsive protein whose function depends on many factors, including intracellular localization. While TG2 is predominantly cytosolic, it migrates in and out of the nucleus (depending on cell type) in response to injury to influence gene expression [34,37,46,47,59]. Our lab and others have shown that TG2’s regulation of gene expression can be mediated through its nuclear interactions, where it may act as a protein scaffold to bring together transcription factors and chromatin regulatory complexes or as an enzyme to catalyze post-translational modifications of epigenetic proteins such as histones [21,22,29,38,40,41,47,57,58,59,60]. In astrocytes, TG2 is primarily associated with transcriptional repression, and given the aforementioned functional effects of astrocytic TG2 deletion, we hypothesize that nuclear interactions by TG2 attenuates the upregulation of neurosupportive signaling pathways in astrocytes during stress.

In line with our hypothesis, inhibition of astrocytic TG2 function by VA4 replicates the neurosupportive effects of TG2−/− astrocytes. VA4 binds to TG2’s catalytic domain, blocks TG2’s catalytic activity, and locks TG2 in the open conformation, which may prevent scaffolding interactions that are dependent on its closed conformation [23]. VA4 therefore putatively limits TG2’s ability to interact with a range of nuclear proteins, so we tested whether VA4 alters the interaction of TG2 and Zbtb7a, a transcription factor that regulates chromatin accessibility and that we previously identified as a TG2 nuclear interactor [15,17,61]. We confirmed that VA4 treatment of astrocytes significantly decreased TG2 binding to Zbtbt7a, suggesting that TG2 requires at least its catalytic domain and/or its closed conformation to bind Zbtb7a. Notably, previous data from our lab using overexpression constructs of mutant TG2 with deficient catalytic and GTP-binding ability suggest that TG2 can still regulate gene expression independent of its catalytic activity and GTP binding [59]. To what degree TG2’s open and closed conformations contribute to its ability to bind to and scaffold between nuclear proteins, and more specifically, which protein domains are important for these interactions, remain open questions.

Interestingly, both TG2 and Zbtb7a can bind to proteins within the Sin3a complex, a transcriptional repressor complex important for gene regulation in response to environmental stress [15,63,65,66]. Sin3a complex proteins associate with the histone deacetylases HDAC1 and HDAC2 to decrease chromatin accessibility [64]. Therefore, we measured acetylated histone levels, markers of chromatin accessibility and gene transcription, in WT and TG2−/− astrocytes. We specifically measured the levels of H3K9ac, as this histone modification is well-established as a mediator of increased chromatin accessibility and transcriptional activity [68]. Both TG2 deletion and VA4 treatment significantly increased the level of H3K9ac in astrocytes. These findings agree with previous RNA sequencing results showing gene upregulations in injured spinal cord tissue from mice with astrocytic TG2 deletion, and a predominant upregulation of genes in cultured TG2−/− astrocytes compared to wild type controls [14,17].

Consistent with our previous RNA sequencing data, our present proteomic analysis of astrocyte cultures shows that TG2 deletion leads to a predominant protein upregulation. Interestingly, however, this trend was not observed in the VA4-treated vs. vehicle-treated WT astrocyte culture comparison, which shows a much wider range of differentially regulated proteins with no clear preference for up- or downregulation. This difference may be partly explained by the acute nature of inhibiting TG2 function relative to TG2−/− astrocytes, but this also suggests that VA4 may have off-target binding effects that need to be further investigated. Strikingly, however, both the TG2−/− and VA4 proteomic comparisons showed strong enrichments in lipid metabolic pathways, consistent with our RNA-seq data of injured spinal cord tissue from astrocytic TG2−/− mice [14]. Within the lipid metabolic pathway, proteins involved in lipid binding and transport (e.g., APOE, FABP7, CPT1a) were consistently upregulated. These changes may be key to the neurosupportive effects of astrocytic TG2 deletion, considering that a rapidly developing frontier in astrocyte biology demonstrates that neurons under stress, and even in in vivo physiological conditions, depend on lipid trafficking and astrocytic lipid metabolism for metabolic homeostasis and controlling oxidative stress [70,71,72,73]. Additionally, following our hypothesis that TG2 acts primarily in an epigenetic role to control astrocytic injury responses, we also analyzed transcription factor pathway enrichments in our proteomic data, which surprisingly showed top enrichments in NFE2L2, or NRF2, pathways. NRF2 is a stress-responsive transcription factor and has a master regulatory role in antioxidant production, among other cell-protective roles relevant to CNS injury [74], and NRF2 serves as an important mediator for astrocytic response to neuronal stress [75,76,77]. Future studies will explore the potential functional differences in lipid and antioxidant metabolism in TG2−/− astrocytes and the importance of these changes for neuron survival in stress conditions.

The transcriptional and proteomic changes from deletion or inhibition of TG2 appear to be dynamic and perhaps very context-dependent so that, across different replicates and analyses, the same genes or proteins are not always differentially expressed, but our data show that the enriched pathways across datasets are remarkably consistent. However, it is worth mentioning that in a re-analysis of our previous in vitro RNA-seq data from TG2−/− astrocytes, we found a unique top enrichment in genes associated with subunits of polycomb repressor complex 2 (PRC2) (i.e., Suz12, EZH2) (Appendix A), which is a transcriptionally repressive chromatin modifying complex that may play a role in TG2-mediated transcriptional repression [78]. A more recent run of RNA sequencing also replicated this enrichment (see Appendix A). Although this same enrichment category was not found in our proteomic data, it is interesting to note that there were consistently significant protein enrichments in H3K27me3 categories, which are histone marks only known to be catalyzed by PRC2 [78]. Across our RNA sequencing and proteomic data, including the VA4 dataset, PRC2 and/or H3K27me3 categories were enriched primarily in downregulated targets. Conversely, in our proteomic data of TG2−/− and VA4-treated astrocytes, pathways associated with Zbtb7a and H3K4me3, a permissive histone mark [79], were consistently enriched in upregulated proteins. Further work is needed to better understand the associations of these bivalent histone marks with up- or downregulated molecular pathways in the absence of TG2 and to determine how TG2 and Zbtb7a may regulate these modifications.

The TG2-Zbtb7a-Sin3a interaction represents a novel and potentially important mechanism that contributes to stress-induced gene regulation by TG2, likely among other contributing players (e.g., PRC2) [15,65]. The Sin3a complex is an important regulator of gene expression during stress, particularly in hypoxic stress [37,66,67]. Indeed, it regulates the vast majority of the transcriptional response to hypoxia and is necessary for a complete response [66]. Our data suggest that TG2 facilitates histone deacetylation, but as TG2 lacks a canonical DNA binding motif, we hypothesize that this effect is mediated through TG2 partnering with DNA-binding transcription factors, like Zbtb7a, and with proteins capable of recruiting a histone de-acetylase, like members of the Sin3a complex. With TG2 deletion or inhibition in astrocytes, repressive epigenetic complexes like Sin3a may have reduced occupancy and activity near stress-induced neurosupportive genes and therefore these genes can be upregulated. Further studies are needed to determine how TG2 interactions with Zbtb7a and Sin3a contribute to TG2’s transcriptionally repressive effect.

## 5. Conclusions

Figure 6 summarizes the proposed mechanism by which TG2 may mediate the neurosupportive status of reactive astrocytes. We found that VA4 inhibition of TG2 disrupts its binding to Zbtb7a, a transcription factor we previously identified as a TG2 nuclear interactor. Additionally, TG2 deletion and inhibition lead to increased transcriptionally permissive histone acetylation, and as both Zbtb7a and TG2 can interact with the HDAC-associated complex, Sin3a, we propose that these interactions facilitate repressive histone deacetylation. A TG2-Zbtb7a-Sin3a mechanism may partly explain the predominant upregulation of gene and protein expression in TG2−/− astrocytes. Further, we propose that this differential gene regulation leads to a more neurosupportive reactive astrocytic phenotype, and our findings indicate that altered lipid and antioxidant pathways largely contribute to these phenotypic changes. Overall, our findings further distinguish TG2 as a multi-faceted transcriptional regulator and an important control point for phenotypic changes in astrocytes responding to stress.

## Figures and Tables

**Figure 1 biomolecules-14-01594-f001:**
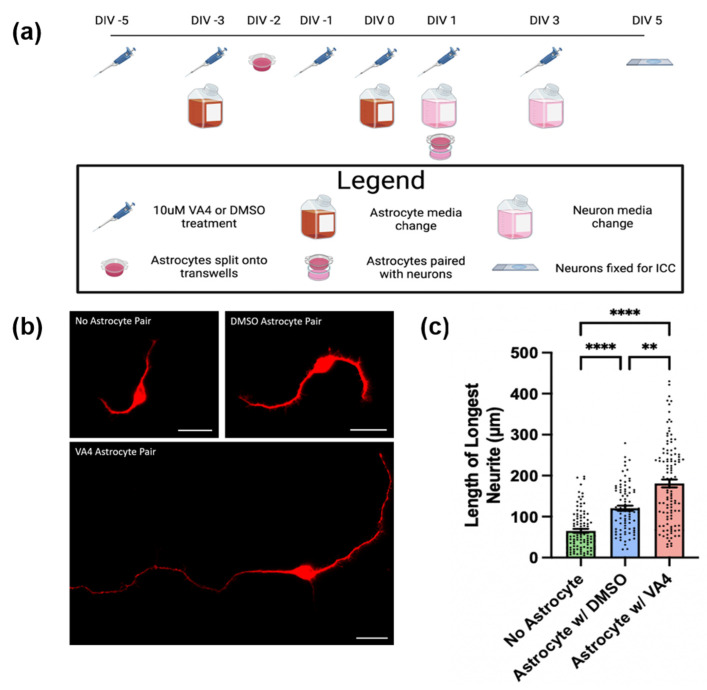
VA4 inhibition of TG2 facilitates the ability of WT astrocytes to promote neurite outgrowth on a chondroitin sulfate proteoglycan (CSPG) growth-inhibitory matrix. (**a**) Schematic of neurite outgrowth experimental paradigm including VA4 treatment. Created with BioRender.com. (**b**) Representative images of MAP2 staining of neurons without paired astrocytes, neurons paired with vehicle (DMSO)-treated astrocytes, or neurons paired with VA4-treated astrocytes (scale bar = 20 µm). (**c**) Quantification of length of longest neurite for neuron experimental groups on CSPG matrix. Shown as mean and SEM (n = 86–111 neurons per group from two biological replicates, Kruskal Wallis test ** *p* < 0.01, **** *p* < 0.0001).

**Figure 2 biomolecules-14-01594-f002:**
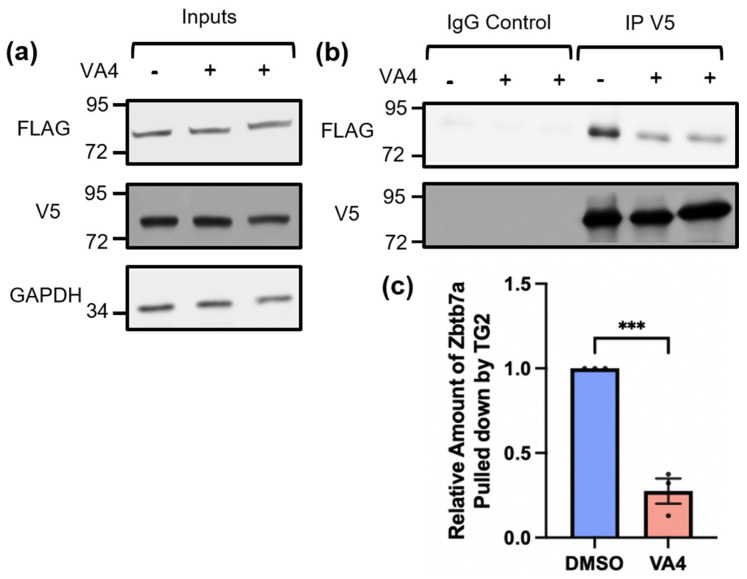
Irreversible inhibition of TG2 with the drug VA4 reduces interaction between TG2 and Zbtb7a. (**a**) Input controls of V5-TG2 and FLAG-Zbtb7a transfected into HEK 293TN cells treated with VA4 or vehicle control (DMSO). (**b**) Immunoprecipitation of V5-TG2 pulls down less FLAG-Zbtb7a in cell lysates that were treated with VA4. In (**a**,**b**) the position at which molecular weight markers (kDa) migrated is indicated at the left of the immunoblots. (**c**) Quantification of the amount of FLAG-Zbtb7a pulled down normalized to the amount of V5-TG2 immunoprecipitated. Treatment of cells with VA4 resulted in a significant reduction in the Zbtb7a co-immunoprecipitated with TG2 compared to DMSO control. Shown as mean and SEM (n = 5 samples per condition from 4 independent biological replicates, unpaired *t*-test *** *p* < 0.001). Original images can be found in Appendix A.

**Figure 3 biomolecules-14-01594-f003:**
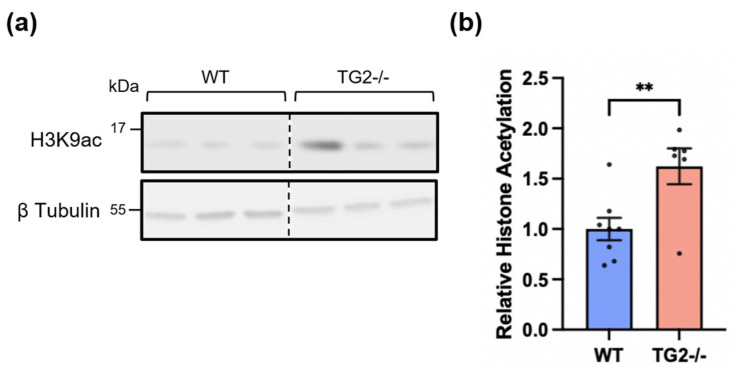
TG2−/− astrocytes have significantly greater acetylation of histone H3 at lysine residue 9 (H3K9ac) compared to WT astrocytes. (**a**) Representative Western blot of WT and TG2−/− astrocyte lysates probed for Histone H3 acetylated at lysine residue 9 (H3K9ac). The position at which molecular weight markers (kDa) migrated is indicated at the left of the immunoblots. (**b**) Quantification of H3K9ac levels in WT and TG2−/− astrocytes. TG2−/− astrocytes showed significantly greater acetylation of H3K9 than WT astrocytes (~60%). Shown as mean and SEM (n = 1–4 samples per condition from 4 independent biological replicates, unpaired *t*-test ** *p* < 0.01). Original images can be found in Appendix A.

**Figure 4 biomolecules-14-01594-f004:**
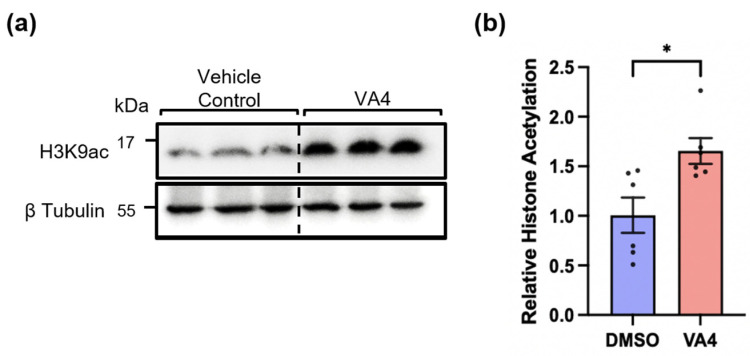
WT astrocytes treated with VA4 show significantly greater H3K9 acetylation compared to DMSO vehicle control treated WT astrocytes. (**a**) Western blot of 10 μM VA4- and DMSO-treated WT astrocyte lysates probed for H3K9ac. The position at which molecular weight markers (kDa) migrated is indicated at the left of the immunoblots. (**b**) Quantification of H3K9ac levels showed significantly greater acetylation in VA4-treated WT astrocytes compared to DMSO-treated WT astrocytes (~65%). Shown as mean and SEM (n = 3 samples per condition from 2 independent biological replicates, unpaired *t*-test * *p* < 0.01). Original images can be found in Appendix A.

**Figure 5 biomolecules-14-01594-f005:**
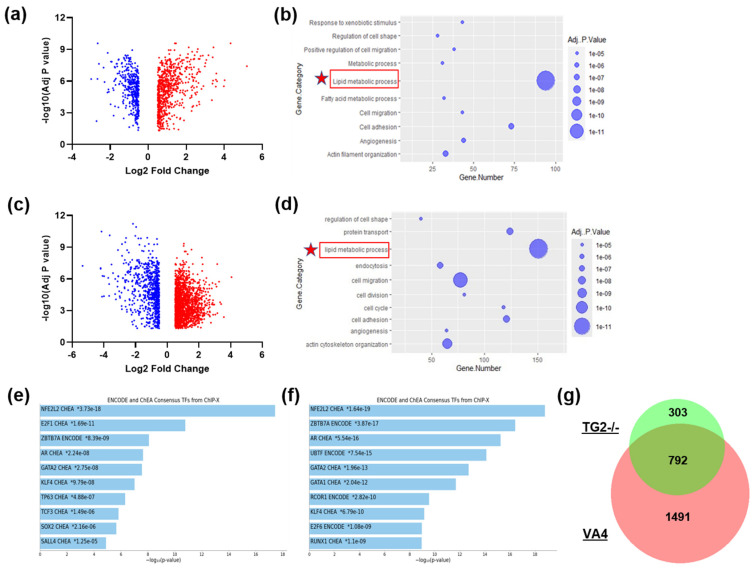
TG2−/− and VA4-treated astrocytes share significant alterations in proteins associated with lipid metabolic and antioxidant pathways. (**a**) Volcano plot of significant differentially regulated proteins comparing TG2−/− to WT astrocyte cultures (n = 5 samples per condition from 1 biological replicate) with thresholds set for log2 fold changes at ±0.5 and for adjusted *p* value < 0.05, and (**b**) DAVID GO Biological Process analysis of up- and downregulated proteins. (**c**) Volcano plot of significant differentially regulated proteins comparing VA4-treated to DMSO-treated WT astrocyte cultures (n = 6 samples per condition from 2 independent biological replicates) with thresholds set as above, and (**d**) DAVID GO Biological Process analysis of up- and downregulated proteins. (**e**,**f**) Enrichr transcription factor enrichment of up- and downregulated proteins in (**e**) TG2−/− astrocytes (**f**) VA4-treated astrocytes (Fisher Exact Test, * *p* < 0.05). (**g**) Venn diagram showing overlap of significant differentially regulated proteins in both TG2−/− and VA4 datasets.

**Figure 6 biomolecules-14-01594-f006:**
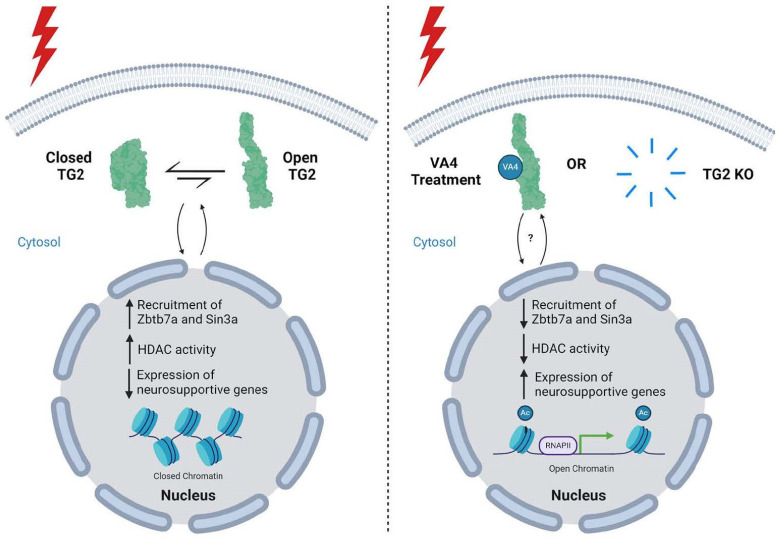
Proposed mechanism by which TG2 modulates the neuroprotective phenotype of reactive astrocytes. In stressed astrocytes, TG2 is able to move in and out of the nucleus. While in the nucleus, TG2 can interact with Zbtb7a and Sin3a, resulting in increased histone deacetylase (HDAC) activity and neurosupportive gene repression. If TG2 is deleted from the astrocytes, or the astrocytes are treated with VA4, the recruitment of Zbtb7a/Sin3a/HDAC to the DNA is diminished, resulting in de-repression of neurosupportive genes leading to a more neurosupportive phenotype. The “?” indicates that the effect of VA4 treatment on the nuclear/cytosolic localization of TG2 is unclear. Created with BioRender.com.

## Data Availability

The original contributions presented in this study are included in the article/Appendix A. Further inquiries can be directed to the corresponding author.

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
