# Peer review of "Pharmacological Inhibition of Astrocytic Transglutaminase 2 Facilitates the Expression of a Neurosupportive Astrocyte Reactive Phenotype in Association with Increased Histone Acetylation"

_biomolecules, 2024, doi:10.3390/biom14121594_

Round 1
Reviewer 1 Report
Comments and Suggestions for Authors
The article "Pharmacological inhibition of astrocytic transglutaminase 2 facilitates the expression of a neurosupportive astrocyte reactive phenotype in association with increased histone acetylation" examines how inhibition of TG2 in astrocytes may potentially favor a neuroprotective phenotypic modification in association with protection of nerve cells under stress conditions. In mouse experiments, the inhibitory effect of TG2 through the VA4 inhibitor was shown to reduce the interactions of TG2 with the transcription factor Zbtb7a, which allows for enhanced histone acetylation and enhanced expression of genes that support nerve cells. Furthermore, suppression of TG2 could lead to pronounced changes in lipid and antioxidant metabolism, which may further enhance the neuroprotective properties of astrocytes.
In my opinion, the article is important because it provides a mechanism by which astrocytes can be pharmacologically modified to support neuronal regeneration and improve functional outcomes after CNS injury. In my opinion, the article could be published if some important issues are addressed:
Analysis of Results and Insight into the Effects of TG2: A more extensive description of how the TG2-Zbtb7a interaction affects histone acetylation and thus the expression of specific genes or proteins involved in neural cell survival is recommended.
Role of Histone Acetylation: Note that the increase in H3K9ac (acetylation on histone H3 at the K9 position) may be even more oriented towards the activation of genes that promote neuronal function and survival, especially in situations of injury.
Changes in Antioxidant Metabolism: Changes in the antioxidant metabolism of astrocytes can enhance their resistance to stress.
Suggestions for Future Research: It is recommended to add a short section discussing possible directions for future research, such as investigations of the effect of VA4 in long-term models of CNS injury, as well as in human cells and studies of the role of TG2 in diseases such as multiple sclerosis plaque or Parkinson's disease.
Similarity of the text to other publications: Paraphrase the sentences using different words and different syntactic structure, and do this especially in cases where the same language is repeated from previous studies. Build on the ideas of other sources of information, but express them in a different way to maintain the authenticity of the text.
Reviewer 2 Report
Comments and Suggestions for Authors
The manuscript entitled “Pharmacological inhibition of astrocytic transglutaminase 2 facilitates the expression of a neurosupportive astrocyte reactive phenotype in association with increased histone acetylation” by Thomas Delgado et al,. in an interesting article about the multiple functions of transglutaminase 2. The manuscript can be accepted after a minor revision. In particular, remove the word “selectively” in the Introduction paragraph (page 2, lane 68).
Comments on the Quality of English LanguageMinor revision.
Reviewer 3 Report
Comments and Suggestions for Authors
The paper written by Delgado et al. investigates the effect of a conformational transglutaminase 2 (TG2) inhibitor, VA4, on the expression of a neurosupportive astrocyte phenotype. Based on their previous results, and present neurite outgrowth and coimmunoprecipitation studies, the authors indicate that TG2 keeps the transcription factor Zbtb7a in the cytosol, but when V4 binds to it, the interaction is disrupted, and Zbtb7a translocates to the nucleus, where it promotes H3k9acetylation.
While some of the data support the hypothesis, some key experiments are missing.
Fig.1c What happens to the neurite outgrowth if VA4 is added to TG2-/- astrocytes?
Fig.2. How is Zbtb7a nuclear translocation altered in TG2+/+ and TG2-/- astrocytes following V4a addition? No data is shown.
Fig.3-4. How is H3k9ac is altered if VA4 is added to TG2-/- astrocytes?
These experiments would make it clear, which effects of VA4 are TG2-independent, whether VA4 can promote neurite outgrowth in the absence of TG2, and in addition, VA4-bound TG2 might also have additional effects, and would demonstrate the key point, that VA4 can regulate Zbtb7a nuclear translocation in a TG-dependent or independent manner.
In the Discussion or in the results we do not have an information what changes in the lipid metabolism mean following VA4 treatment. Do the authors have any idea how the two cell types communicate? If they feel via lipid signaling molecules, was lipid analysis made in the cell culture medium?
Minor point
I think in lane 88/ Chaper 3.5, the sentence would be more correct this way:
While as compared to wild-type astrocytes VA4 astrocytes show a wider range of differentially
regulated proteins than the TG2-/- astrocytes, TG2-/- astrocytes share the majority, about 70%, of their differentially regulated proteins with VA4-treated astrocytes.
Round 2
Reviewer 1 Report
Comments and Suggestions for Authors
The authors made all corrections. The revised version is completely agreeable to me, for this reason, I recommend the publication of the manuscript.
Reviewer 3 Report
Comments and Suggestions for Authors
With the changes and explanations and the added new informations the paper became more clear, and interesting for the proper audience.